# An exceptionally flexible hydrogen-bonded organic framework with large-scale void regulation and adaptive guest accommodation abilities

Qiuyi Huang[1,3], Wenlang Li[1,3], Zhu Mao[1], Lunjun Qu[1], Yang Li[2], Hao Zhang[2], Tao Yu[1], Zhiyong Yang[1], Juan Zhao[1], Yi Zhang[1], Matthew P. Aldred[1] & Zhenguo Chi [1]

Flexible hydrogen-bonded organic frameworks (FHOFs) are quite rare but promising for applications in separation, sensing and host-guest chemistry. They are difficult to stabilize, making their constructions a major challenge. Here, a flexible HOF (named 8PN) with permanent porosity has been successfully constructed. Nine single crystals of 8PN with different pore structures are obtained, achieving a large-scale void regulation from 4.4% to 33.2% of total cell volume. In response to external stimuli, multimode reversible structural transformations of 8PN accompanied by changes in luminescence properties have been realized. Furthermore, a series of high-quality co-crystals containing guests of varying shapes, sizes, aggregation states and even amounts are obtained, showing that 8PN can adapt to different guests by regulating the molecular conformations and assembling forms of its building blocks. The unexpected flexibility of 8PN makes it a promising material for enriching the applications of existing porous materials.

[1] PCFM Lab, GDHPPC Lab, Guangdong Engineering Technology Research Center for High-performance Organic and Polymer Photoelectric Functional Films, State Key Laboratory of OEMT, School of Chemistry, Sun Yat-sen University, 510275 Guangzhou, China. [2] Instrumental Analysis and Research Center (IARC), Sun Yat-Sen University, 510275 Guangzhou, China. [3]These authors contributed equally: Qiuyi Huang, Wenlang Li. Correspondence and requests for materials should be addressed to Y.Z. (email: ceszy@mail.sysu.edu.cn)

Porous materials, such as metal-organic frameworks (MOFs) and covalent organic frameworks (COFs), have been extensively investigated over the past few decades, which are potentially attractive for various applications, including gas storage/separation, catalysis, chemical sensors, drug delivery, and so on[1–6]. Among them, flexible MOFs (FMOFs) have generated considerable research interest, which exhibit reversible structural transformations when responding to external stimuli[7]. Compared with robust porous materials, the dynamic behaviors of FMOFs make them very promising materials for separation and sensing applications[8]. For the majority of the investigated FMOFs, conformational changes of the organic ligands in different environments are responsible for the framework flexibility[9]. In general, when incorporating some flexible organic linkers in a porous material, the ability that adopting different conformations under different external stimuli can be reasonably transferred from the organic moiety to the whole framework[10,11]. Therefore, porous frameworks are expected to be more flexible when they are constructed solely from flexible organic molecules. However, in contrast to the well-developed FMOFs, entirely organic flexible porous frameworks are extremely rare.

Recently, an emerging class of porous materials termed as hydrogen-bonded organic frameworks (HOFs), which are constructed from pure organic or metal-containing organic building blocks through hydrogen-bonding interactions, have attracted increasing attention[12]. Although there are many pioneering studies[13–15], the development status of HOFs lags behind that of MOFs. The main reason for this is that the vast majority of HOFs will collapse upon removal of the solvent molecules in the pores. It was only until recently (2010) that the demonstration of HOFs exhibiting permanent porosity was realized[16,17]. Due to the numerous inherent advantages, such as low cost, easy purification, facile regeneration by recrystallization, potential water tolerance, and highthermal stability[18,19], HOFs are quite promising for gas storage/separation[20–24], proton conduction[25,26], molecular recognition[27,28], optical applications[29], and so on. Although weak hydrogen bonds are responsible for the instability of HOFs[30], compared with strong covalent bonds and coordination bonds, they are rather more flexible[31]. The utilization of soft hydrogen bonds can also be beneficial to the framework flexibility of the porous material. It is worth noting that in order to construct flexible HOFs (FHOFs), the organic molecules should be moderately flexible, as opposed to HOFs that can be quite robust and built using only rigid building blocks[32]. Through a combination of soft hydrogen bonding and flexible organic building blocks, developing purely organic flexible porous frameworks is feasible and indeed logical. A flexible HOF could potentially realize multimode structural transformations in response to different kinds of external stimuli, including both physical and chemical stimuli. This is due to its flexibility, which can hardly be achieved in a single FMOF, making itself a superior candidate for sensing, host–guest chemistry, and hopefully some other fields. However, only very few HOFs have demonstrated both permanent porosity and intriguing dynamic behaviors.

Herein, we report an exceptionally flexible HOF, henceforth denoted as 8PN, with permanent porosity. By regulating the molecular conformations and assembling forms, a large-scale void regulation and multimode reversible structural transformations are realized in 8PN. Due to the outstanding flexibility, 8PN can also adapt to guests of varying shapes, sizes, aggregation states, and even amounts.

## Results

**Design strategy for the organic building block.** The organic building block of 8PN is 1,1,2,2-tetrakis(4′-nitro-[1,1′-biphenyl]-4-yl) ethane (TPE-4pn), which is based on the tetraphenylethylene (TPE) backbone (Fig. 1). TPE is an archetypal aggregation-induced emission (AIE) fluorophore[33], providing the porous framework with extraordinary solid-state luminescence properties. Moreover, as a flexible organic molecule, TPE is expected to exhibit conformational changes upon different stimuli by modulating the dihedral angles between planes of phenyl rings and the ethylene core. Besides, additional four phenyl rings are attached to the *para*-positions of the TPE moiety, resulting in more flexibility. All the softness of the molecular building block resulting from phenyl rotations will reasonably be transferred to the final porous framework. In addition, the nitro group that serves as a hydrogen bond acceptor, should be able to link together neighboring organic building blocks through intermolecular hydrogen bonds. The changes of the hydrogen bond distances upon external stimuli can also contribute to the dynamic behaviors of 8PN. Consequently, the flexible HOF 8PN can be successfully constructed by combining the flexible nature of the TPE-4pn molecules with the flexibility of the hydrogen bonds.

**Single-crystal structures of 8PN.** Eight kinds of single crystals of 8PN (with solvent inclusion) can be readily obtained by the solvent evaporation method. Single-crystal X-ray diffraction reveals that the flexible TPE-4pn is able to adopt different conformations depending on different solvent conditions, resulting in frameworks with various pore structures (Fig. 2b; Supplementary Figs. 2–9). Most commonly, each quadrangle-shaped pore in 8PN is assembled by four nonporous TPE-4pn molecules, including 8PN-ACT, 8PN-DMF, 8PN-EA, 8PN-2ACT, 8PN-TCM, and 8PN-THF, through four pairs of intermolecular C–H–O hydrogen bonds between the aromatic C–H groups and oxygen atoms from the nitro-group linkages. Each TPE-4pn molecule is connected with four neighboring ones by hydrogen bonds, generating a supramolecular single layer. Then a two-dimensional (2D) double layer is formed by two adjacent single layers with a small misalignment via interlayer hydrogen bonds, whilst the overall structures of the two single layers in the double layer exhibit central symmetry with each other. The double layer is further stacked together through numerous hydrogen-bonding interactions, building up a three-dimensional (3D) porous supramolecular structure with S-shaped channels (Fig. 2c). These six frameworks can be further categorized into three types, named

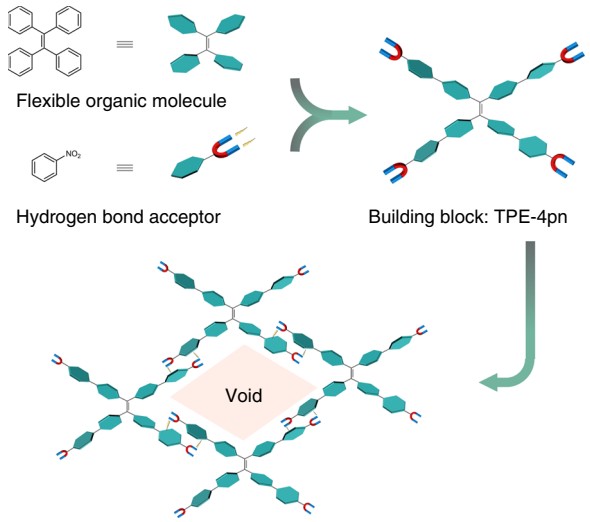

**Fig. 1** The design strategy for 8PN. The flexible framework is constructed from TPE-4pn through hydrogen-bonding interactions

Flexible organic molecule

Hydrogen bond acceptor

Building block: TPE-4pn

Void

Flexible hydrogen-bonded organic framework: 8PN

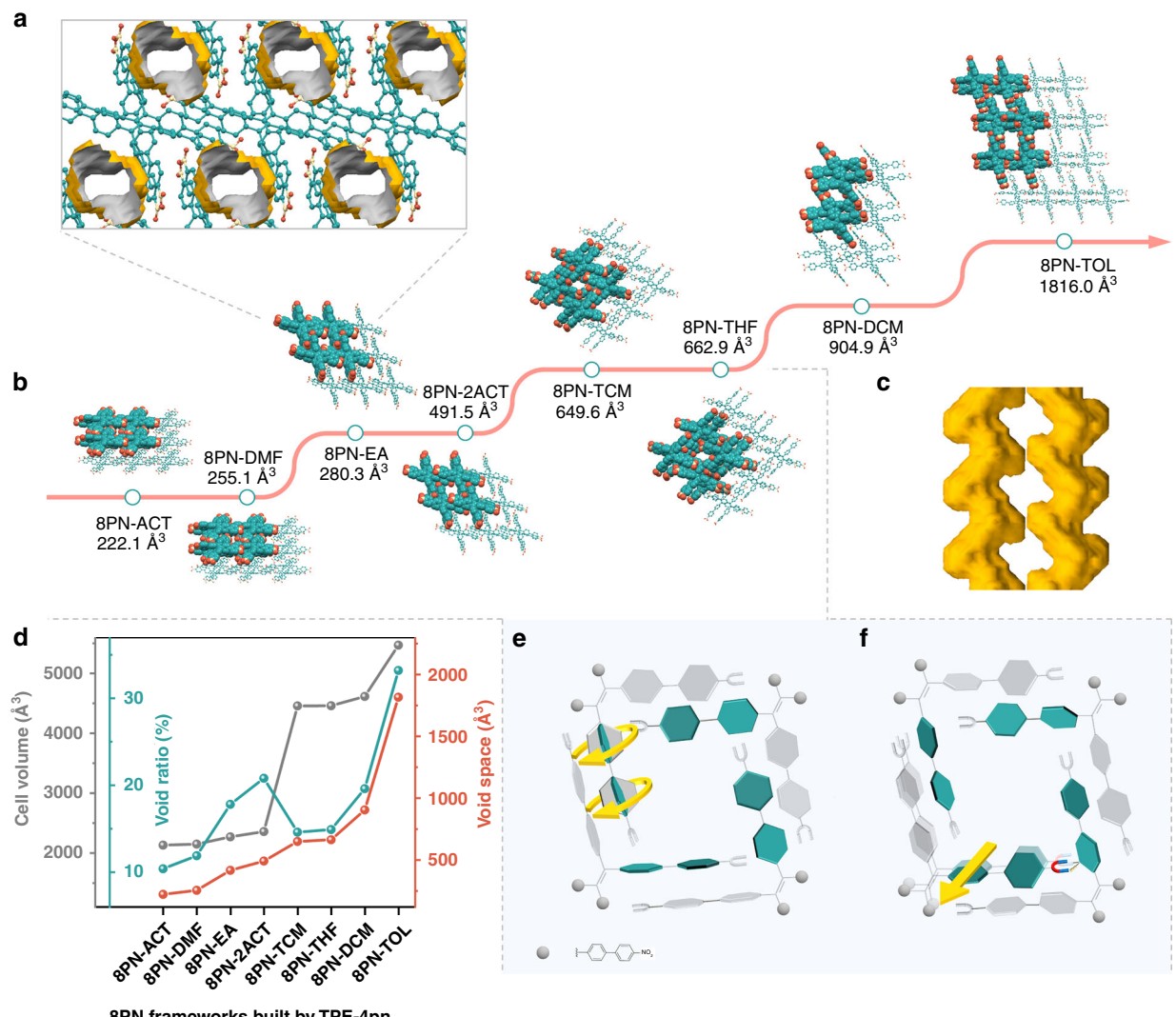

**Fig. 2** Investigation of single-crystal structures of 8PN. **a** Packing diagram of 8PN-EA along the [010] direction, with the solvent-accessible void space visualized by gray/yellow (inner/outer) curved planes generated with a probe of 1.2 Å. Color code: green, C; yellow, N; Orange, O. H atoms are omitted for clarity. **b** Single-crystal X-ray structures of eight 8PN frameworks with varying void space. H atoms and solvent molecules in voids are omitted for clarity. Abbreviations for different solvents: ACT acetone, DMF *N,N*-dimethylformamide, EA ethyl acetate, TCM chloroform, THF tetrahydrofuran, DCM dichloromethane, TOL toluene. **c** Visualization of the S-shaped channel surface of 8PN-THF. **d** Void spaces, cell volumes, void ratios of the eight 8PN frameworks. **e** Illustration of large-scale regulation of the pore size by dihedral angles among different types of 8PN-200, 8PN-400, and 8PN-600. **f** Illustration of small-scale regulation of the pore size by hydrogen bond distances in the two frameworks within the same type, that is 8PN-ACT and 8PN-DMF in 8PN-200 type, 8PN-EA, and 8PN-2ACT in 8PN-400 type or 8PN-TCM and 8PN-THF in 8PN-600 type

8PN-200, 8PN-400, and 8PN-600, according to their unit cell parameters and void spaces. In general, regarding the three types, the larger dihedral angles between the planes of phenyl rings (defined as planes A, B, C, D, E, F, G, and H) and the ethylene core (defined as planes X and Y) leads to the larger pore size (Fig. 2e; Supplementary Fig. 12 and Supplementary Table 4). In the 8PN-200 type, although dihedral angles between A and X (A^X) of TPE-4pn are relatively large, dihedral angles between B and X (B^X) are as small as 29.53° and 29.12°, so that planes B partially occupy the space of the pores. The sizes of pores in the frameworks of 8PN-ACT and 8PN-DMF are $6.621 \times 9.892$ Å$^2$ and $6.709 \times 10.253$ Å$^2$, respectively, and the solvent-accessible void spaces are 222.1 Å$^3$ (10.4% of the total cell volume) and 255.1 Å$^3$ (11.9% of the total cell volume) per cell, respectively, as estimated by Platon in the absence of all solvent molecules[34]. Dihedral angles A^X, B^X, C^Y, and D^Y of TPE-4pn in the type of 8PN-400 are modest, leading to medium-sized pores with

void spaces of 380.3 Å$^3$ (16.8% of the total cell volume) and 491.5 Å$^3$ (20.8% of the total cell volume) in 8PN-EA (Fig. 2a) and 8PN-2ACT, respectively. With regards to the 8PN-600 type, all the dihedral angles A^X and C^Y are larger than 55°, which effectively enlarges the space of the pores. As a result, among the three types, 8PN-TCM and 8PN-THF possess large pore sizes of $7.547 \times 13.106$ Å$^2$ and $7.704 \times 13.110$ Å$^2$, with void spaces of 649.6 Å$^3$ (14.6% of the total cell volume) and 662.9 Å$^3$ (14.9% of the total cell volume). In contrast to the relatively large-scale regulation of the pore size among different types by dihedral angles, the distances of hydrogen bonds between the four adjacent TPE-4pn molecules that form a pore in the framework, play an auxiliary role. Changes of C–H–O hydrogen bond distances can adjust the pore size within a minor range in the same framework type (Fig. 2f; Supplementary Fig. 13 and Supplementary Table 5). For example, in the 8PN-200 type, distances of C–H–O hydrogen bonds in 8PN-DMF are 2.622, 2.526, 2.922, and 2.723 Å, which

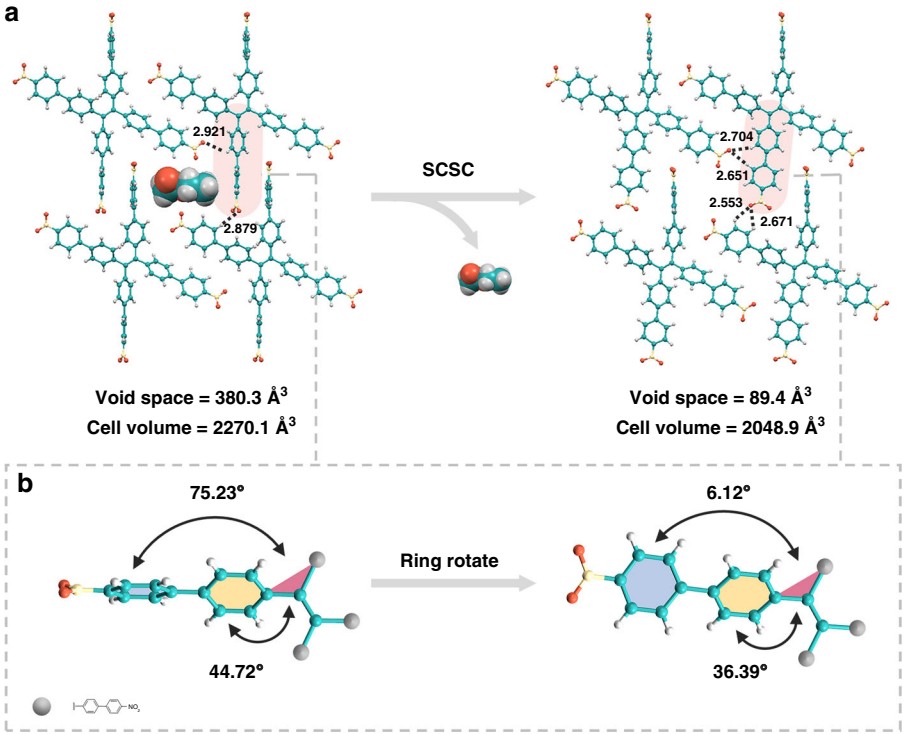

**Fig. 3** Single-crystal-to-single-crystal (SCSC) transformation determined by the single-crystal X-ray diffraction. **a** Structural transformations of the large pore (LP) form in 8PN-EA to the narrow pore (NP) form in 8PN-Heated accompanied by removal of solvent molecules. **b** Illustration of the significant changes in the dihedral angles between the plane of the ethylene core and the planes of two different phenyl rings in one arm of TPE-4pn in 8PN-EA and 8PN-Heated due to phenyl ring-rotation

are longer compared with the corresponding locations in 8PN-ACT (2.612, 2.478, 2.835, and 2.626 Å). Relatively long hydrogen bond distances lead to the four adjacent TPE-4pn molecules away from each other, making the pore size slightly larger, which also holds true for types 8PN-400 and 8PN-600. In addition to the aforementioned four TPE-4pn molecules constructing a pore, two TPE-4pn molecules can also create a pore in the dichloromethane/n-hexane system, building up a framework 8PN-DCM with estimated void space of 904.9 Å$^3$ (19.6% of the total cell volume). Surprisingly, in the toluene/methanol solvent system, an exceptional framework 8PN-TOL with two types of pores that share two TPE-4pn molecules as one of their side emerges. Platon calculations on 8PN-TOL indicate a void space of 1816.0 Å$^3$ (33.2% of the total cell volume), higher than that found in any other frameworks of 8PN. Therefore, through changes of molecular conformations and assembling forms, a large adjustable range of void space can be achieved in the supramolecular framework 8PN (Fig. 2d).

**Framework flexibility of 8PN.** To determine the permanent porosity of these 8PN frameworks above, the $CO_2$ gas adsorption measurements at 195 K and $N_2$ gas adsorption measurements at 77 K were carried out. According to the thermogravimetric analyses (TGA) data (Supplementary Fig. 14), solvent molecules in the voids of various frameworks of 8PN can be released by heating to 160 °C for 8PN-DMF and 120 °C for others. The crystal structures of the desolvated samples of 8PN-TCM and 8PN-THF are identical to their pristine solvated-type single-crystal structures, and 8PN-DCM and 8PN-TOL just exhibit only mild contraction of the crystal structure[35] (Supplementary Fig. 15b–e). However, the frameworks of 8PN- ACT, 8PN-DMF, and 8PN-EA are transformed into one equivalent crystal

structure that has not been found in 8PN after losing their solvent molecules (Supplementary 16a–f). In order to get that single-crystal structure, a single crystal of 8PN-EA was heated carefully until all the EA molecules were removed to yield another single crystal named 8PN-Heated. Upon the solvent removal, the single crystallinity of 8PN can be retained. When cooled to ambient temperature, the crystalline framework of 8PN-Heated is still stable and the single-crystal structure of 8PN-Heated can be successfully obtained and proved to be the equivalent crystal structure as mentioned vide supra (Supplementary Fig. 15a). During the guest removal process, a single-crystal-to-single-crystal (SCSC) transformation takes place (Fig. 3a). After EA removal, two dihedral angles between the plane of the ethylene core and the planes of two different phenyl rings in one arm of TPE-4pn change from 44.72° and 75.23° to 36.39° and 6.12° due to rotations of the phenyl rings (Fig. 3b), leading to the transformation of the large pore (LP) form to the narrow pore (NP) form. Therefore, the aforementioned general rule between dihedral angles and pore sizes is further clearly confirmed. 8PN-Heated exhibits the smallest pore size of 6.232 × 9.624 Å$^2$ and void space of 89.4 Å$^3$ (4.4% of the total cell volume), featuring only 90% of the initial cell volume of 8PN-EA.

The $CO_2$ gas adsorption isotherm of 8PN-Heated clearly indicates its microporous nature (Supplementary Fig. 17). Remarkably, for all the activated frameworks of 8PN, large hysteretic sorption behaviors are observed during the $CO_2$ adsorption/desorption process (Fig. 4a; Supplementary Figs. 18–20). This is not common in HOFs and indicates the framework flexibility of 8PN[36]. All the activated frameworks of 8PN exhibit uptake of $CO_2$, but no adsorption of $N_2$ at 77 K. The higher quadrupole moment of $CO_2$ can enhance the interactions between $CO_2$ and the framework, resulting in the increase of $CO_2$ binding[20]. The permanent porosity of 8PN, proven by the above $CO_2$ gas adsorption results, may be related to the

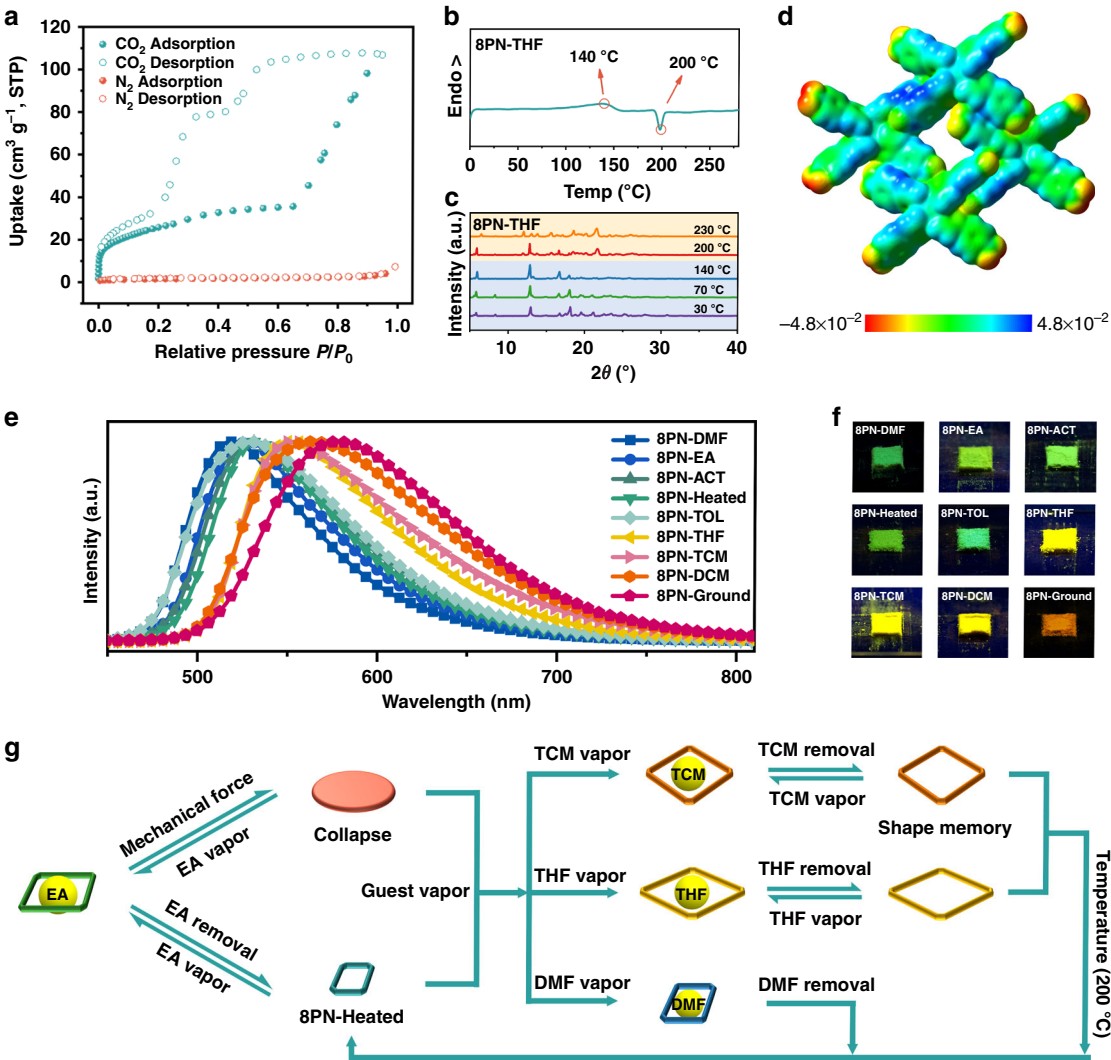

**Fig. 4** Characterization and multimode reversible transformations of porous structures of 8PN. **a** $CO_2$ (195 K) and $N_2$ (77 K) adsorption/desorption isotherms for 8PN-THF. **b** Differential scanning calorimetry (DSC) curve of 8PN-THF. **c** Variable-temperature powder X-ray diffraction (PXRD) patterns of 8PN-THF. **d** Electrostatic potential (ESP) diagram of the pore constructed by four TPE-4pn molecules in 8PN-THF indicating the nitro groups inside the pore present the less electron density (in yellow color) than the nitro groups outside the pore (in red color). The potential energy range is $-4.8 \times 10^{-2}$ to $4.8 \times 10^{-2}$. **e** Emission spectra of 8PN excited at 365 nm. **f** Photographs of 8PN taken under UV-light irradiation (365 nm). **g** Illustration of the multimode reversible structural transformations realized in 8PN including transitions between various porous phases as well as transitions between nonporous and porous states in response to different kinds of external stimuli. Source data are provided as a Source Data file

multiple hydrogen-bonding interactions in the supramolecular structures, which can be reflected in the electrostatic potential (ESP) diagram (Fig. 4d; Supplementary Fig. 23). The reduced electron density on the nitro groups inside the pore is caused by the formation of four pairs of symmetric C–H–O hydrogen bonds between the aromatic C–H groups and O atoms from the nitro groups inside the pore[37]. The porous crystalline state can be maintained by these hydrogen bonds.

The flexibility of 8PN does not only manifest itself in the $CO_2$ gas adsorption/desorption process but also in its multimode structural transformations (Fig. 4g). As previously mentioned, in response to guest removal, the single-crystal structure of 8PN-EA is able to convert from the LP form to the NP form of 8PN-Heated. Also, the structural transformation between NP and LP can also occur via adsorption of EA molecules. Furthermore, the obtained crystals of 8PN-Heated can also be reversibly converted to crystals of other frameworks of 8PN by exposure to the corresponding solvent vapors (Supplementary Fig. 21). The as-obtained framework of 8PN-DMF containing DMF molecules in its pores can also lose the guest

molecules by heating up to 160 °C to produce 8PN-Heated, realizing the reversible LP-to-NP transformation as well. Similar transformations can also be observed for 8PN-ACT. Therefore, for 8PN-Heated, the adsorption processes of EA, DMF, and ACT are dominated by the kinetically controlled flexibility (KCF)[38]. However, frameworks of 8PN-TCM and 8PN-THF can release solvent molecules from their pores by heating to 120 °C to produce the corresponding frameworks with almost unchanged empty pores. Similarly, after removing the solvent molecules from the voids, 8PN-DCM and 8PN-TOL just exhibit mild structural contraction. The adsorption behaviors of TCM, THF, DCM, and TOL for 8PN-Heated can be assigned to thermodynamically controlled flexibility (TCF)[38]. The relatively slower non-radiative decay ($k_{nr}$) rates of 8PN-TCM, 8PN-THF, 8PN-DCM, and 8PN-TOL probably resulted from slow molecular vibrations greatly contribute to the stabilities of their frameworks, maintaining unchanged pores after solvents removal (Supplementary Table 6). Even more surprisingly, when the empty frameworks of 8PN-TCM, 8PN-THF, 8PN-DCM, and 8PN-TOL are further heated to 200 °C, they are all transformed to 8PN-Heated (Fig. 4b, c;

Supplementary Fig. 16g–l). Once cooled down to the room temperature, the structure of the as-obtained 8PN-Heated can be maintained. The original framework of 8PN-TCM or others can be regenerated upon application of the corresponding vapor. Therefore, in the presence of two different kinds of external stimuli, including temperature change and guest change, the shape-memory effect has been realized between 8PN-Heated and 8PN-TCM or 8PN-THF[39]. Moreover, two different empty shape-memory pores of 8PN-TCM and 8PN-THF can be converted to each other, achieving a multimode shape-memory behavior, which can be extremely attractive for memory device applications. In addition, mechanical force can structurally transform 8PN, in which after grinding 8PN-EA loses its crystallinity and converts to the amorphous state, generating a nonporous sample named 8PN-Ground. 8PN-Ground can also recover the crystallinity of 8PN-EA smoothly on being exposed to EA vapor (Supplementary Fig. 22). Thus, in response to different external stimuli, 8PN can realize multimode reversible structural transformations, including structural transitions between various porous phases and nonporous-to-porous transitions. That is, 8PN richly exhibits the so-called breathing effect in so many different forms.

TPE is an archetypal AIE-active molecule and, therefore, exhibits interesting optical properties. Different forms of 8PN exhibit unstructured and broad emission profiles ranging from 518 nm to 580 nm (Fig. 4e), with lifetimes from 0.93 ns to 4.4 ns (Supplementary Fig. 25). Among them, 8PN-THF shows the strongest fluorescence with an impressive photoluminescence quantum yield ($\Phi_F$) of 0.55. According to time-dependent density functional theory (TD-DFT) calculations, the fluorescence emission of 8PN can be attributed to typical charge transfer (CT) effect (Supplementary Fig. 24). As a result, when responding to different external stimuli, 8PN not only shows multimode transformations in its structure, but also, at the same time, exhibits abundant changes in luminescence properties including emission color, lifetime, and brightness. The visualization structural transformations are easily detected, making 8PN also promising for chemical sensors, data recording, biomedical imaging, light-emitting devices, and other smart materials.

**Host–guest co-crystals.** The exceptional flexibility of 8PN has also been applied to host–guest chemistry especially to accommodate solid-state guests, which is seldom investigated in conventional porous frameworks. It is worth noting that high-quality single crystals of porous materials containing guests are generally hard to be obtained. But single crystals are so valuable that characteristics like linker disorder, guest arrangement, guest amounts, and host–guest interactions can hardly be determined with atomic precision without single crystals. As a small organic molecule, it is easy for TPE-4pn to grow high-quality single crystals. By employing the liquid-assisted grinding (LAG) method[40], five high-quality co-crystals of 8PN and different guests suitable for single-crystal X-ray diffractions have been successfully prepared (Fig. 5; Supplementary Figs 26–30). For the host–guest co-crystals of 8PN-NDP and 8PN-NPNA, the two solid-state guests, namely diphenylamine (NDP) and N-phenylnaphthalen-2-amine (NPNA), are quite similar in shape but not in size. The flexible 8PN can encapsulate the two guests, respectively, by the adjustment of the molecular conformations of TPE-4pn in the frameworks. Numerous intermolecular hydrogen-bonding interactions are established to stabilize the co-crystal structures (Supplementary Fig. 31a, b and Supplementary Table 7), which are impossible to be observed when only the polycrystalline samples are available. Furthermore, the guests that can be encapsulated by 8PN are not limited by the shape of the guest as well. Besides the two twisted nonplanar guests mentioned above, other guests such as the highly planar pyrene (PY) and the

relatively slender 3-hexylthiophene (SC6) can also be encapsulated by 8PN. By changing the assembling forms and the molecular conformations, co-crystals of 8PN-PY and 8PN-SC6 can be readily obtained. Surprisingly, when the ratio of TPE-4pn to pyrene used in the co-crystal preparation changes from 1:1 to 1:3, another co-crystal named 8PN-3PY is formed. It can be clearly determined from the single-crystal structure of 8PN-3PY that this co-crystal contains one TPE-4pn molecule and three pyrene molecules in its unit cell, in other words, a 1:3 host–guest co-crystal structure can be achieved. Host–guest interactions contributing to the stability of these structures can also be observed in their single crystals. Therefore, 8PN exhibits unexpected flexibility to adapt to guests of varying shapes, sizes, aggregation states, and even amounts. 8PN can serve as a multipurpose host, which can accommodate many special-purpose guests with excellent physical and chemical properties to create host–guest co-crystal materials. These special co-crystal materials are very likely to possess some unexpected properties and functions originating from the specific guests and could have critical potential applications in some special fields, which common porous materials can hardly realize by employing only themselves. Therefore, as a multifunctional platform, the flexible 8PN is quite promising for enriching the applications of existing porous materials. The employment of a flexible host in these host–guest co-crystals also provide a valuable strategy for the preparation of conventional co-crystals. The types of co-crystals can be richer, and the preparations could be easier by selecting a flexible host as one component of the co-crystals. All of these are a target for our future follow-up studies.

## Discussion

In summary, we have successfully assembled an exceptionally flexible porous hydrogen-bonded organic framework (8PN) with permanent porosity from 1,1,2,2-tetrakis(4′-nitro-[1,1′-biphenyl]-4-yl)ethane. Through regulating the assembling forms and molecular conformations, nine single crystals of 8PN with different pore structures are obtained, achieving a large-scale regulation of void space from 89.4 Å$^3$ (4.4% of the total cell volume) to 1816.0 Å$^3$ (33.2% of the total cell volume). In response to different external stimuli, including guest, temperature, and mechanical pressure changes, multimode reversible structural transformations among large pore forms, narrow pore forms, and nonporous forms can be realized. 8PN also exhibits changes in luminescence properties upon external stimuli, making the structural transformations visual. Furthermore, five high-quality co-crystals are obtained, demonstrating that the flexible 8PN can adapt to guests of varying shapes, sizes, aggregation states, and even amounts. The unexpected flexibility of 8PN makes it attractive for enriching the application fields of existing porous materials. The employment of a flexible host in these host–guest co-crystals should also provide valuable advice for co-crystal preparations.

## Methods

**Synthesis of the organic building block TPE-4pn.** 1,1,2,2-tetrakis(4-bromophenyl)ethene (2.00 g, 3.10 mmol) and (4-nitrophenyl)boronic acid (3.09 g, 18.5 mmol) were dissolved in THF (50.00 mL). Subsequently, 2 M K$_2$CO$_3$ aqueous solution (6.00 mL) and Aliquat 336 (0.50 mL) were added. The mixture was stirred for 10 min under nitrogen atmosphere at room temperature. Then Pd(PPh$_3$)$_4$ (0.50 mg) was added and the mixture was stirred at 80 °C for 24 h. After cooling down to room temperature, the crude product was concentrated and purified by silica gel column chromatography with dichloromethane/n-hexane (v/v = 1:1) as the eluent. Compound TPE-4pn was obtained as a yellow solid in 68% yield (1.70 g). The structural characterization is provided in the Supplementary Information section.

**Preparation of the single crystals of 8PN.** The single crystals of 8PN-ACT and 8PN-DMF suitable for X-ray diffraction analysis were grown, respectively, by slow solvent evaporation of a saturated solution of TPE-4pn in acetone or N,N-dimethylformamide at room temperature for 1 week. For 8PN-EA, TPE-4pn (5.00 mg,

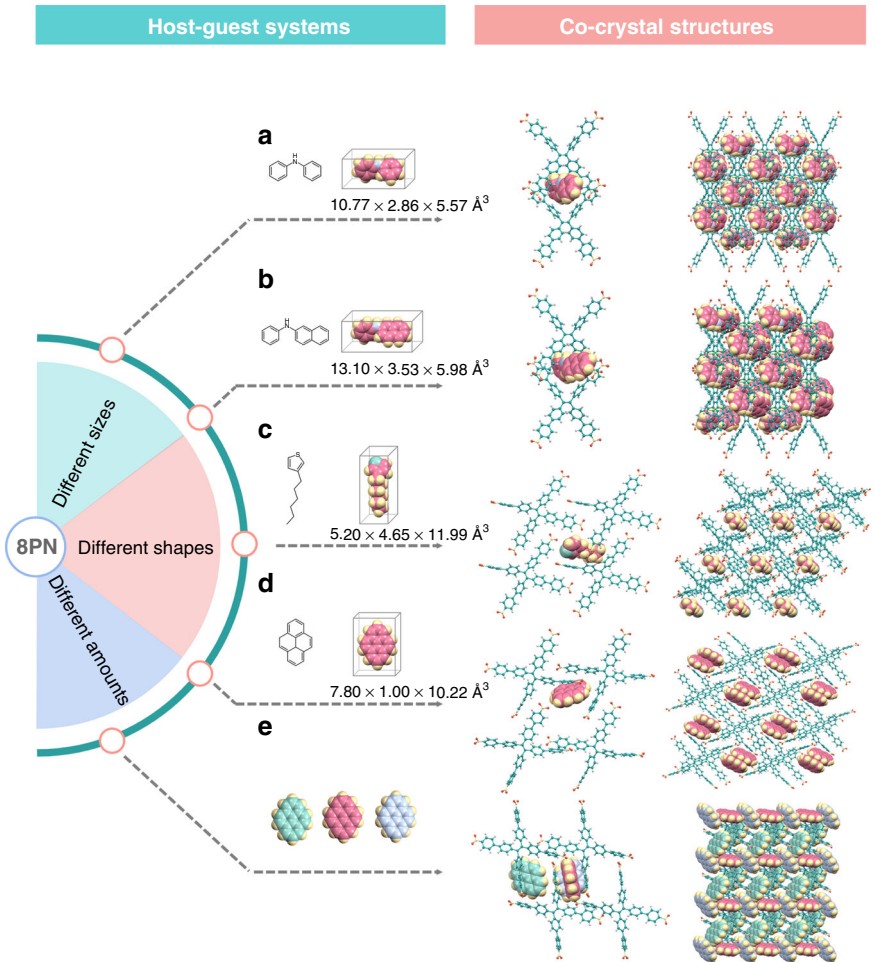

**Fig. 5** Summary of the adaptive accommodations of different guests by 8PN in various host−guest co-crystal structures. **a** Molecular size of diphenylamine and single-crystal X-ray structure of 8PN-NDP. **b** Molecular size of *N*-phenylnaphthalen-2-amine and single-crystal X-ray structure of 8PN-NPNA. **c** Molecular size of 3-hexylthiophene and single-crystal X-ray structure of 8PN-SC6. **d** Molecular size of pyrene and single-crystal X-ray structure of 8PN-PY. **e** single-crystal X-ray structure of 8PN-3PY. Color code for TPE-4pn: green, C; yellow, N; orange, O; gray, H. Color code for guests: pink, C; yellow, H; blue, N; light green, S, except that C atoms of the three pyrene molecules in the unit cell of 8PN-3PY are colored pink, light green, and light blue, respectively, for clarity

0.006 mmol) was dissolved in ethyl acetate (5.00 mL) and methanol (1.00 mL). The single crystals of 8PN-EA suitable for X-ray diffraction analysis were obtained by slow solvent evaporation of the resulting solution of TPE-4pn at room temperature for one week. The preparations of single crystals of 8PN-2ACT, 8PN-THF, and 8PN-TOL are similar to that of 8PN-EA but with acetone, tetrahydrofuran, or toluene instead of ethyl acetate, respectively. The preparations of single crystals of 8PN-TCM and 8PN-DCM are similar to that of 8PN-EA, but with chloroform or dichloromethane instead of ethyl acetate and *n*-hexane instead of methanol.

**Preparations of host–guest co-crystals**. The organic building block TPE-4pn served as the host, while diphenylamine, N-phenylnaphthalen-2-amine, 3-hexylthiophene, and pyrene were chosen as the guests. By employing a typical liquid-assisted grinding (LAG) method, 50 μL of dichloromethane was added to a solid mixture (20.00 mg) with a 1:1 molar ratio of the host and the guest. The mixture was ground for 15 min. By using different guests, respectively, the experiments described above gave rise to three host–guest co-crystals 8PN-NDP, 8PN-NPNA, and 8PN-PY. These LAG-produced co-crystal powders were, respectively, dissolved in dichloromethane (20.00 mL) and methanol (3.00 mL). Their single crystals suitable for X-ray diffraction analysis were obtained, respectively, by slow solvent evaporation of the resulting solutions of these co-crystal powders at room temperature for a week. A procedure similar to that of 8PN-PY was followed but with a 1:3 molar ratio of the host and the guest instead of the 1:1 molar ratio to produce single crystals of 8PN-3PY. As the guest 3-hexylthiophene is liquid at room temperature, 3-hexylthiophene (0.50 mL) was directly added into a saturated dichloromethane/methanol (v/v = 1:1) solution (4.00 mL) of TPE-4pn. By slow solvent evaporation at room temperature for a week, single crystals of 8PN-SC6 suitable for X-ray diffraction analysis were obtained.

## Data availability
The X-ray crystallographic coordinates for structures reported in this study have been deposited at the Cambridge Crystallographic Data Centre (CCDC), under deposition numbers CCDC 1891754, 8PN-ACT; CCDC 1891755, 8PN-DMF; CCDC 1891756, 8PN-EA; CCDC 1891757, 8PN-2ACT; CCDC 1891766, 8PN-TCM; CCDC 1891758, 8PN-THF; CCDC 1891759, 8PN-DCM; CCDC 1891760, 8PN-TOL; CCDC 1891761, 8PN-Heated; CCDC 1891762, 8PN-NDP; CCDC 1891763, 8PN-NPNA; CCDC 1891764, 8PN-SC6; CCDC 1891765, 8PN-PY; CCDC 1891767, 8PN-3PY. These data can be obtained free of charge from The Cambridge Crystallographic Data Centre via www.ccdc.cam.ac.uk/data_request/cif. All other relevant data are available in the Supplementary Information, as well as from the corresponding authors upon reasonable request. The source data underlying Fig. 4a–c, e and Supplementary Figs. 17–20 are provided as a Source Data file. The data files are available at https://figshare.com/articles/Source_Data_xlsx/8050358.

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

## Acknowledgements

This work was financially supported by the National Natural Science Foundation of China (NSFC: 51733010, 21672267, 51703253 and 51873239), Science and Technology Planning Project of Guangdong (2015B090913003 and 2015B090915003), China Post-doctoral Science Foundation (2017M620395), the Leading Scientific, Technical and Innovation Talents of Guangdong Special Support Program (2016TX03C295), and the Fundamental Research Funds for the Central Universities.

## Author contributions

Q.H., W.L., Y.Z. and Z.C. conceived and designed the experiments. Q.H., W.L., Z.M., L.Q., Y.L. and H.Z. performed the experiments. T.Y., Y.Z., J.Z. and other authors were all involved in the analyses and interpretation of data. Q.H. and W.L. wrote the paper with the help of Y.Z., Z.C. and M.P.A.

## Additional information

**Competing interests:** The authors declare no competing interests.

