## [Peer Review File · Nature Communications]

Reviewers' comments:

Reviewer #1 (Remarks to the Author):

This is a very interesting piece of work and has the merit to be published in Nature Communications. The main claim of this work is HOFs, not MOFs, so it is not necessary to cite too many refs on MOFs while lacks quite a lot refs on HOFs. Please check the recent review article (Chem. Soc. Rev., 2019, 48, 1362-1389) to have a whole picture in this topic and choose the refs accordingly. Fig. 2d, the illustration might be not so accurate, because it needs be based on the fact that all unit cells are the same for all structures. % of total cell volumes will be much more reliable to compare.

Reviewer #2 (Remarks to the Author):

Solid-state porous architectures based on non-covalent interactions have attracted increasing interest due to their unique advantages and potential applications. Compared to MOFs, porous hydrogen-bonded organic frameworks (HOFs) are readily purified and recovered via simple recrystallization. However, due to lacking of sufficiently ability to orientate self-aggregation of building motifs in predictable manners, rational design and preparation of porous HOFs are still challenging. Zhang and coworkers reported that HOFs can be constructed from TPE-4pn molecules and still keep the porosity when solvents are removed. They tried different solvents for the crystallization to vary pore sizes. It is very impressive for the permanent porosity via the CO₂ adsorption-desorption isotherm. More interestingly, their HOFs exhibit the light-emitting property. In a word, I would like to highly recommend this work to be published in Nature Communications.

Reviewer #3 (Remarks to the Author):

In this manuscript, Zhang et al. reported the assembly of one porous hydrogen-bonded organic framework (HOF) fabricated from TPE molecule. Different external stimuli led to the regulation of void space of as-synthesized HOF 8PN. Additionally, five co-crystals with different guest molecules were also obtained. Although the work seems to be interesting, the manuscript is written in a very simplistic way and with poor explanations. The as-mentioned "permanent porosity" can not be supported by the low CO₂ adsorption capacity and absence of N₂ adsorption/desorption data. Moreover, similar flexible TPE-based HOFs with SC-SC features have also been reported before (J. Am. Chem. Soc. 2015, 137, 9963). Therefore, I do not think it is worthy of publishing in Nature Communications, while a more specialized journal relating to supramolecular chemistry would be more appropriate.

Point-by-point response to the reviewers' comments:

(Reviewers' comments and suggestions: in black; Responses to the comments and suggestions: in blue)

Reviewer #1 (Remarks to the Author):

This is a very interesting piece of work and has the merit to be published in Nature Communications.

Response: We would like to thank you for your positive comments and valuable suggestions to improve our manuscript.

1. The main claim of this work is HOFs, not MOFs, so it is not necessary to cite too many refs on MOFs while lacks quite a lot refs on HOFs. Please check the recent review article (Chem. Soc. Rev., 2019, 48, 1362-1389) to have a whole picture in this topic and choose the refs accordingly.

Response: As suggested by the reviewer, we have modified the references. References on MOFs have been reduced. According to the recent review article (Chem. Soc. Rev., 2019, **48**, 1362-1389), some necessary descriptions of HOFs and references on HOFs have been added in the revised manuscript. Our changes made to the revised manuscript are highlighted in yellow as follows:

On page 2-3 in the revised manuscript:

Novel porous materials, such as metal-organic frameworks (MOFs) and covalent organic frameworks (COFs), have been extensively investigated over the past few decades, which are potentially attractive for various applications, including gas storage/separation, catalysis, chemical sensors, drug delivery, and so on¹⁻⁶. Amongst

them, flexible MOFs (FMOFs) have generated considerable research interest, which exhibit reversible structural transformations when responding to external stimuli⁷. Compared with robust porous materials, the dynamic behaviors of FMOFs make them very promising materials for separation and sensing applications⁸. For the majority of the investigated FMOFs, conformational changes of the organic ligands in different environments are responsible for the framework flexibility⁹. In general, when incorporating some flexible organic linkers in a porous material, the ability that adopting different conformations under different external stimuli can be reasonably transferred from the organic moiety to the whole framework¹⁰⁻¹¹. Therefore, porous frameworks are expected to be more flexible when they are constructed solely from flexible organic molecules. However, in contrast to the well-developed FMOFs, entirely organic flexible porous frameworks are extremely rare.

Recently, an emerging class of porous materials termed as hydrogen-bonded organic frameworks (HOFs), which are constructed from pure organic or metal-containing organic building blocks through hydrogen-bonding interactions, have attracted increasing attention¹². Although there are many pioneering studies¹³⁻¹⁵, the development status of HOFs lags behind that of MOFs. The main reason for this is that the vast majority of HOFs will collapse upon removal of the solvent molecules in the pores. It was only until recently (2010) that the demonstration of HOFs exhibiting permanent porosity was realized¹⁶⁻¹⁷. Due to the numerous inherent advantages, such as low cost, easy purification, facile regeneration by recrystallization, potential water

tolerance and high thermal stability¹⁸⁻¹⁹, HOFs are quite promising for gas storage/separation²⁰⁻²⁴, proton conduction²⁵⁻²⁶, molecular recognition²⁷⁻²⁸, optical applications²⁹, and so on. Although weak hydrogen bonds are responsible for the instability of HOFs³⁰, compared with strong covalent bonds and coordination bonds, they are rather more flexible³¹.

On page 22-25 in the revised manuscript:

2. Li, J.-R., Sculley, J. & Zhou, H.-C. Metal–organic frameworks for separations. *Chem. Rev.* **112**, 869-932 (2012).
3. Ding, S.-Y. *et al.* Construction of covalent organic framework for catalysis: Pd/COF-LZU1 in Suzuki-Miyaura coupling reaction. *J. Am. Chem. Soc.* **133**, 19816-19822 (2011).
4. Cui, Y., Yue, Y., Qian, G. & Chen, B. Luminescent functional metal-organic frameworks. *Chem. Rev.* **112**, 1126-1162 (2012).
5. Dalapati, S., Jin, E., Addicoat, M., Heine, T. & Jiang, D. Highly emissive covalent organic frameworks. *J. Am. Chem. Soc.* **138**, 5797-5800 (2016).
6. Rocca, J. D., Liu, D. & Lin, W. Nanoscale metal-organic frameworks for biomedical imaging and drug delivery. *Acc. Chem. Res.* **44**, 957-968 (2011).
7. Horike, S., Shimomura, S. & Kitagawa, S. Soft porous crystals. *Nat. Chem.* **1**, 695-704 (2009).
8. Chang, Z., Yang, D.-H., Xu, J., Hu, T.-L. & Bu, X.-H. Flexible metal–organic frameworks: recent advances and potential applications. *Adv. Mater.* **27**, 5432-5441 (2015).

9. Henke, S., Schneemann, A., Wütscher, A. & Fischer, R. A. Directing the breathing behavior of pillared-layered metal–organic frameworks via a systematic library of functionalized linkers bearing flexible substituents. *J. Am. Chem. Soc.* **134**, 9464-9474 (2012).
10. Chatterjee, B. *et al.* Self-assembly of flexible supramolecular metallacyclic ensembles: structures and adsorption properties of their nanoporous crystalline frameworks. *J. Am. Chem. Soc.* **126**, 10645-10656 (2004).
11. Dong, Y.-B. *et al.* Temperature-dependent synthesis of metal-organic frameworks based on a flexible tetradentate ligand with bidirectional coordination donors. *J. Am. Chem. Soc.* **129**, 4520-4521 (2007).
12. Lin, R.-B. *et al.* Multifunctional porous hydrogen-bonded organic framework materials. *Chem. Soc. Rev.* **48**, 1362-1389 (2019).
13. Simard, M., Su, D. & Wuest, J. D. Use of hydrogen bonds to control molecular aggregation. Self-assembly of three-dimensional networks with large chambers. *J. Am. Chem. Soc.* **113**, 4696-4698 (1991).
14. Brunet, P., Simard, M. & Wuest, J. D. Molecular tectonics. Porous hydrogen-bonded networks with unprecedented structural integrity. *J. Am. Chem. Soc.* **119**, 2737-2738 (1997).
15. Fournier, J.-H., Maris, T., Wuest, J. D., Guo, W. & Galoppini, E. Molecular tectonics. Use of the hydrogen bonding of boronic acids to direct supramolecular construction. *J. Am. Chem. Soc.* **125**, 1002-1006 (2003).
16. He, Y., Xiang, S. & Chen, B. A microporous hydrogen-bonded organic framework

- for highly selective C₂H₂/C₂H₄ separation at ambient temperature. *J. Am. Chem. Soc.* **133**, 14570-14573 (2011).
17. Yang, W. *et al.* Exceptional thermal stability in a supramolecular organic framework: porosity and gas storage. *J. Am. Chem. Soc.* **132**, 14457-14469 (2010).
18. Hu, F. *et al.* An ultrastable and easily regenerated hydrogen-bonded organic molecular framework with permanent porosity. *Angew. Chem. Int. Ed.* **56**, 2101-2104 (2017).
19. Cai, S. *et al.* Hydrogen-bonded organic aromatic frameworks for ultralong phosphorescence by intralayer π - π interactions. *Angew. Chem. Int. Ed.* **57**, 4005-4009 (2018).
20. Luo, X.-Z. *et al.* A microporous hydrogen-bonded organic framework: exceptional stability and highly selective adsorption of gas and liquid. *J. Am. Chem. Soc.* **135**, 11684-11687 (2013).
21. Wang, H. *et al.* A flexible microporous hydrogen-bonded organic framework for gas sorption and separation. *J. Am. Chem. Soc.* **137**, 9963-9970 (2015).
22. Li, P. *et al.* A rod-packing microporous hydrogen-bonded organic framework for highly selective separation of C₂H₂/CO₂ at room temperature. *Angew. Chem. Int. Ed.* **54**, 574-577 (2015).
23. Yang, W. *et al.* Highly interpenetrated robust microporous hydrogen-bonded organic framework for gas separation. *Cryst. Growth Des.* **17**, 6132-6137 (2017).
24. Bao, Z. *et al.* Fine tuning and specific binding sites with a porous

- hydrogen-bonded metal-complex framework for gas selective separations. *J. Am. Chem. Soc.* **140**, 4596-4603 (2018).
25. Yang, W. *et al.* Microporous diaminotriazine-decorated porphyrin-based hydrogen-bonded organic framework: permanent porosity and proton conduction. *Cryst. Growth Des.* **16**, 5831-5835 (2016).
26. Karmakar, A. *et al.* Hydrogen-bonded organic frameworks (HOFs): a new class of porous crystalline proton-conducting materials. *Angew. Chem. Int. Ed.* **55**, 10667-10671 (2016).
27. Li, P. *et al.* A homochiral microporous hydrogen-bonded organic framework for highly enantioselective separation of secondary alcohols. *J. Am. Chem. Soc.* **136**, 547-549 (2014).
28. Wang, H. *et al.* A microporous hydrogen-bonded organic framework with amine sites for selective recognition of small molecules. *J. Mater. Chem. A* **5**, 8292-8296 (2017).
29. Wang, H. *et al.* Two solvent-induced porous hydrogen-bonded organic frameworks: solvent effects on structures and functionalities. *Chem. Commun.* **53**, 11150-11153 (2017).
30. Hisaki, I. *et al.* Docking strategy to construct thermostable, single-crystalline, hydrogen-bonded organic framework with high surface area. *Angew. Chem. Int. Ed.* **57**, 12650-12655 (2018).

2. Fig. 2d, the illustration might be not so accurate, because it needs be based on the fact that all unit cells are the same for all structures. % of total cell volumes will be much more reliable to compare.

Response: We appreciate the good suggestion from the reviewer. % of total cell volumes (solvent-accessible void ratios) are much more reliable to compare, which can reflect the porosity more accurately and relate to the gas storage capacity to a certain extent. Following the reviewer's useful suggestions, we have replaced the original Fig. 2d with the following figure in the revised manuscript. As depicted in the following figure, a threefold increase in solvent-accessible void ratio from 10.4% to 33.2% is realized by using different solvents in crystallization processes. Moreover, a solvent-accessible void ratio of 4.4% is obtained in the framework of 8PN-Heated. So, on the whole, as shown in Supplementary Fig. 11, a large adjustable range of solvent-accessible void ratio up to 7.5 times can be achieved in response to different external stimuli in the supramolecular framework 8PN. We have added the data of % of total cell volumes when we compare the porosity of different frameworks of 8PN in the revised manuscript.

In addition, for the regulation of void by dihedral angles and hydrogen bond distances, what we want to present is the regulation of the size of each pore in different frameworks. In the six frameworks including 8PN-ACT, 8PN-DMF, 8PN-EA, 8PN-2ACT, 8PN-TCM and 8PN-THF, changes in dihedral angles between the planes of phenyl rings and the ethylene core can lead to the changes in pore size. And changes in C-H \cdots O hydrogen bond distances can also lead to the changes in pore size. In general, the larger dihedral angles and the longer hydrogen bond distances will lead to the larger pore size. This rule is further confirmed through the single-crystal-to-single-crystal (SCSC) transformation from 8PN-EA to 8PN-Heated revealed by single-crystal X-ray diffraction analysis. Moreover, through changes of molecular assembling forms, the pore shape can also be regulated. The sizes and shapes of pores in different frameworks relate to the adaptive guest accommodation ability of 8PN. As a result, as described in the "Host-guest co-crystals" part, due to the tunable pore size and pore shape resulted from the exceptional flexibility of 8PN, the guests of varying shapes, sizes, aggregation states and even amounts can all be encapsulated by 8PN. To clarify, data of pore sizes have been added in the Supplementary Table 3. Related description about the regulation of pore size by dihedral angles

and hydrogen bond distances has also been revised.

Our changes made to the revised manuscript are highlighted in yellow as follows:

On page 1 in the revised manuscript:

Nine single crystals of 8PN with different pore structures were obtained, achieving a large-scale void regulation from 4.4% to 33.2% of total cell volume.

On page 8-9 in the revised manuscript:

Figure 2 | Investigation of single-crystal structures of 8PN. d, Void spaces, cell volumes, void ratios of the eight 8PN frameworks. e, Illustration of large-scale regulation of the pore size by dihedral angles among different types of 8PN-200, 8PN-400 and 8PN-600. f, Illustration of small-scale regulation of the pore size by hydrogen bond distances in the two frameworks within the same type, that is 8PN-ACT and 8PN-DMF in 8PN-200 type, 8PN-EA and 8PN-2ACT in 8PN-400 type or 8PN-TCM and 8PN-THF in 8PN-600 type.

On page 6-8 in the revised manuscript:

In general, regarding the three types, the larger dihedral angles between the planes of phenyl rings (defined as planes A, B, C, D, E, F, G and H) and the ethylene core (defined as planes X and Y) leads to the larger pore size (Fig. 2e and Supplementary Fig. 12 and Table 4). In the 8PN-200 type, although dihedral angles between A and X

(A[^]X) of TPE-4pn are relatively large, dihedral angles between B and X (B[^]X) are as small as 29.53° and 29.12°, so that planes B partially occupy the space of the pores.

The sizes of pores in the frameworks of 8PN-ACT and 8PN-DMF are $6.621 \times 9.892 \text{ \AA}^2$ and $6.709 \times 10.253 \text{ \AA}^2$, respectively, and the solvent-accessible void spaces are 222.1 \AA^3 (10.4% of total cell volume) and 255.1 \AA^3 (11.9% of total cell volume) per cell, respectively, as estimated by Platon in the absence of all solvent molecules³⁴.

Dihedral angles A[^]X, B[^]X, C[^]Y and D[^]Y of TPE-4pn in the type of 8PN-400 are modest, leading to medium-sized pores with void spaces of 380.3 \AA^3 (16.8% of total cell volume) and 491.5 \AA^3 (20.8% of total cell volume) in 8PN-EA (Fig. 2a) and

8PN-2ACT, respectively. With regards to the 8PN-600 type, all the dihedral angles A[^]X and C[^]Y are larger than 55°, which effectively enlarges the space of the pores.

As a result, amongst the three types, 8PN-TCM and 8PN-THF possess large pore sizes of $7.547 \times 13.106 \text{ \AA}^2$ and $7.704 \times 13.110 \text{ \AA}^2$, with void spaces of 649.6 \AA^3 (14.6% of total cell volume) and 662.9 \AA^3 (14.9% of total cell volume). In contrast to

the relatively large-scale regulation of the pore size amongst different types by dihedral angles, the distances of hydrogen bonds between the four adjacent TPE-4pn molecules that form a pore in the framework, play an auxiliary role. Changes of C-H···O hydrogen bond distances can adjust the pore size within a minor range in the same framework type (Fig. 2f and Supplementary Fig. 13 and Table 5). For example, in the 8PN-200 type, distances of C-H···O hydrogen bonds in 8PN-DMF are 2.622, 2.526, 2.922, 2.723 Å, which are longer compared to the corresponding locations in 8PN-ACT (2.612, 2.478, 2.835, 2.626 Å). Relatively long hydrogen-bond distances

lead to the four adjacent TPE-4pn molecules away from each other, making the pore size slightly larger, which also holds true for types 8PN-400 and 8PN-600. In addition to the aforementioned four TPE-4pn molecules constructing a pore, two TPE-4pn molecules can also create a pore in dichloromethane/*n*-hexane system, building up a framework 8PN-DCM with estimated void space of 904.9 Å³ (19.6% of total cell volume). Surprisingly, in toluene /methanol solvent system, an exceptional framework 8PN-TOL with two types of pores that share two TPE-4pn molecules as one of their side emerges. Platon calculations on 8PN-TOL indicate a void space of 1816.0 Å³ (33.2% of total cell volume), higher than that found in any other frameworks of 8PN. Therefore, through changes of molecular conformations and assembling forms, a large adjustable range of void space can be achieved in the supramolecular framework 8PN (Fig. 2d).

On page 10 in the revised manuscript:

8PN-Heated exhibits the smallest pore size of $6.232 \times 9.624 \text{ \AA}^2$ and void space of 89.4 Å³ (4.4% of total cell volume), featuring only 90% of the initial cell volume of 8PN-EA.

On page S10 in the revised manuscript:

Supplementary Table 3 | Data for voids

HOFs	V (Å ³) ^a	Ratio (%) ^b	Pore size (Å ²) ^c
8PN-Heated	89.4	4.4	6.232 × 9.624
8PN-ACT	222.1	10.4	6.621 × 9.892
8PN-DMF	255.1	11.9	6.709 × 10.253
8PN-EA	380.3	16.8	6.507 × 12.290
8PN-2ACT	491.5	20.8	8.679 × 12.912
8PN-TCM	649.6	14.6	7.547 × 13.106
8PN-THF	662.9	14.9	7.704 × 13.110
8PN-DCM	904.9	19.6	12.666 × 15.023
8PN-TOL (L)			10.638 × 17.606
8PN-TOL (N)	1816.0	33.2	9.143 × 18.582

^a V : Solvent-accessible void space (Å³).

^b Ratio: Void space ratio (%).

^c Pore size: Distance of atom centers including vdW radii (Å²).

Reviewer #2 (Remarks to the Author):

Solid-state porous architectures based on non-covalent interactions have attracted increasing interest due to their unique advantages and potential applications. Compared to MOFs, porous hydrogen-bonded organic frameworks (HOFs) are readily purified and recovered via simple recrystallization. However, due to lacking of sufficiently ability to orientate self-aggregation of building motifs in predictable manners, rational design and preparation of porous HOFs are still challenging. Zhang and coworkers reported that HOFs can be constructed from TPE-4pn molecules and still keep the porosity when solvents are removed. They tried different solvents for the crystallization to vary pore sizes. It is very impressive for the permanent porosity via the CO₂

adsorption-desorption isotherm. More interestingly, their HOFs exhibit the light-emitting property. In a word, I would like to highly recommend this work to be published in Nature Communications.

Response: We would like to thank you very much for your careful reviewing and positive comments on our manuscript. We are very pleased to learn your recognition of our work.

Reviewer #3 (Remarks to the Author):

In this manuscript, Zhang et al. reported the assembly of one porous hydrogen-bonded organic framework (HOF) fabricated from TPE molecule. Different external stimuli led to the regulation of void space of as-synthesized HOF 8PN. Additionally, five co-crystals with different guest molecules were also obtained. Although the work seems to be interesting, the manuscript is written in a very simplistic way and with poor explanations.

Response: We would like to thank you for your suggestive questions and valuable comments to improve our manuscript.

1. The as-mentioned “permanent porosity” can not be supported by the low CO₂ adsorption capacity and absence of N₂ adsorption/desorption data.

Response: We would like to thank you for your helpful reminding and good suggestion. The N₂ gas adsorption measurements are also necessary and important. According to the reviewer’s suggestion, we have done the N₂ adsorption measurements and added the N₂ adsorption/desorption data in Fig. 4a and Supplementary Figs 17-20.

As shown in these figures, all the activated samples of different frameworks of 8PN exhibit uptake of CO₂ at 195 K, but no adsorption of N₂ even at 77 K. Actually, such phenomenon is common in many reported HOFs with permanent porosity and several HOFs can selectively absorb CO₂ rather than N₂. The quadrupole moment of CO₂ ($-1.4 \times 10^{-39} \text{ C}\cdot\text{m}^2$) is higher than that

of N₂ ($4.7 \times 10^{-40} \text{ C}\cdot\text{m}^2$). It has been reported that the high quadrupole moment of CO₂ can enhance the interactions between CO₂ and the framework, resulting in the increase of CO₂ binding. And the small quadrupole moment of N₂ can lead to the correspondingly low uptake (*J. Am. Chem. Soc.*, 2013, **135**, 11684-11687; *Chem. Rev.*, 2012, **112**, 869-932). To make the manuscript more complete, the description about the N₂ adsorption/desorption results has been added in the manuscript.

And at the time, we are sorry for our unclear description about the permanent porosity determined by the CO₂ gas adsorption measurements.

On the one hand, as shown in Supplementary Fig. 17, the CO₂ adsorption capacity of 8PN-Heated is assuredly low, but it is in accordance with the small solvent-accessible void ratio (4.4%). And at the same time, 8PN-Heated is obtained by heating. Both of the thermogravimetric analyses (TGA) data and single-crystal X-ray diffraction (SXRD) analysis can confirm that there is no solvent molecule in the pore of 8PN-Heated, while the void space of 89.4 \AA^3 (4.4% of total cell volume) can still be estimated by Platon and an amount of CO₂ adsorption can still be observed in the CO₂ adsorption/desorption isotherms. So, although the pore in 8PN-Heated is small, it can still be identified as a framework with permanent porosity.

On the other hand, for other frameworks of 8PN, for example, for the activated sample of 8PN-THF, the amount of CO₂ uptake is $106.9 \text{ cm}^3\cdot\text{g}^{-1}$ (195 K, $P/P_0 = 0.95$), as shown in Fig. 4a. In consideration of the not enough large solvent-accessible void ratio (14.9%) compared with that of different HOFs reported in previous literature, it is understandable that the CO₂ adsorption capacity seems not so impressive. But this does not prevent us from determining permanent porosity of 8PN-THF and other frameworks of 8PN. Of course, many HOFs exhibit outstanding CO₂ adsorption capacity. However, in fact, as shown in the following table, by comparing the amount of CO₂ adsorption of different HOFs with permanent porosity reported in the literature, we can find that such an amount of CO₂ adsorption ($106.9 \text{ cm}^3\cdot\text{g}^{-1}$) can actually support the permanent porosity.

Table 1 | CO₂ adsorption data confirming permanent porosity of the corresponding HOF in several literatures

HOFs	CO ₂ uptake (cm ³ ·g ⁻¹)	Reference
HOF-2a	~85 ^a	J. Am. Chem. Soc. , 2014, 136 , 547-549
HOF-6a	~90 ^a	Cryst. Growth Des. , 2016, 16 , 5831-5835.
HOF-7a	~73 ^a	Cryst. Growth Des. , 2015, 15 , 2000-2004
1-2D-apo	95.6 ^b	Angew. Chem. Int. Ed. , 2015, 54 , 3008–3012
Benzotrisimidazole	63 ^b	Chem. Commun. , 2016, 52 , 4991-4994
HOF-GS-10	~43 ^b	Angew. Chem. Int. Ed. , 2016, 55 , 10667-10671
HOF-GS-11	~33 ^b	Angew. Chem. Int. Ed. , 2016, 55 , 10667-10671
HOF-10a	~63 ^a	Chem. Commun. , 2017, 53 , 11150-11153
UTSA-300a	~95 ^b	J. Am. Chem. Soc. , 2017, 139 , 8022–8028
HOF-21a	~98 ^a	J. Am. Chem. Soc. , 2018, 140 , 4596-4603
8PN-Heated	14.7 ^b	Our work
8PN-TCM	100.5 ^b	Our work
8PN-THF	106.9 ^b	Our work
8PN-DCM	91.7 ^b	Our work
8PN-TOL	43.6 ^b	Our work

^a CO₂ adsorption at 196 K.

^b CO₂ adsorption at 195 K.

Following the reviewer's suggestions, our changes made to the revised manuscript are highlighted in yellow as follows:

On page 9 in the revised manuscript:

To determine the permanent porosity of these 8PN frameworks above, the CO₂ gas adsorption measurements at 195 K and N₂ gas adsorption measurements at 77 K were carried out.

On page 11 in the revised manuscript:

All the activated frameworks of 8PN exhibit uptake of CO₂, but no adsorption of N₂ at 77 K. The higher quadrupole moment of CO₂ can enhance the interactions between CO₂ and the framework, resulting in the increase of CO₂ binding²⁰.

On page 14 in the revised manuscript:

Figure 4 | Characterization and multimode reversible transformations of porous structures of 8PN. a, CO₂ (195 K) and N₂ (77 K) adsorption/desorption isotherms for 8PN-THF. Source data are provided as a Source Data file.

On page S3 in the revised manuscript:

CO₂ gas adsorption/desorption isotherms were obtained by a Micromeritics ASAP 2020 surface area analyzer. N₂ gas adsorption/desorption isotherms were obtained by a Micromeritics ASAP 2460 surface area analyzer.

On page S21-S22 in the revised manuscript:

Supplementary Figure 17 | CO₂ (195 K) and N₂ (77 K) adsorption/desorption isotherms for 8PN-Heated. Source data are provided as a Source Data file.

Supplementary Figure 18 | CO₂ (195 K) and N₂ (77 K) adsorption/desorption isotherms for 8PN-TCM. Source data are provided as a Source Data file.

Supplementary Figure 19 | CO₂ (195 K) and N₂ (77 K) adsorption/desorption isotherms for 8PN-DCM. Source data are provided as a Source Data file.

Supplementary Figure 20 | CO₂ (195 K) and N₂ (77 K) adsorption/desorption isotherms for 8PN-TOL. Source data are provided as a Source Data file.

2. Moreover, similar flexible TPE-based HOFs with SC-SC features have also been reported before (*J. Am. Chem. Soc.* 2015, 137, 9963). Therefore, I do not think it is worthy of publishing in *Nature Communications*, while a more specialized journal relating to supramolecular chemistry would be more appropriate.

Response: The article mentioned by the reviewer (*J. Am. Chem. Soc.*, 2015, **137**, 9963-9970) reports a flexible HOF (HOF-5) constructed through numerous strong N–H···N hydrogen bonding interactions. Through reversible structural transformation during solvent removal process, the authors successfully established the structure of the activated sample (HOF-5a) by the powder X-ray diffraction (PXRD) analysis. HOF-5a exhibits high CO₂ adsorption capacity.

In our manuscript, we present a flexible HOF (8PN), which is constructed through relatively weak but flexible C–H···O hydrogen bonding interactions. 8PN can exhibit exceptional framework flexibility. In response to different external stimuli including guest, temperature and mechanical pressure changes, multimode reversible structural transformations including structural transitions between various porous phases and nonporous-to-porous transitions accompanied by changes in luminescence properties have been achieved in 8PN. Thereinto, single-crystal-to-single-crystal (SCSC) transformation can be observed. And a multimode shape-memory behavior between 8PN-Heated, 8PN-TCM and 8PN-THF can also be observed. Such abundant reversible structural transformations involving eight porous frameworks and one nonporous state of 8PN are rarely reported for HOFs.

We are sorry for our unclear description about the SCSC transformation in our manuscript, which affected your judgment on the significance of our work to a certain extent.

In our manuscript, we reported a SCSC transformation from the large pore form of 8PN-EA to the narrow pore form of 8PN-Heated. However, not only the structure of 8PN-EA, but also the structure of 8PN-Heated are obtained directly from the single-crystal X-ray diffraction (SXRD) analysis. The HOF of 8PN can retain its single crystallinity upon the solvent removal due to the exceptional framework flexibility. SXRD has numerous merits. Characteristics like atomic positions, geometric parameters (bond lengths and angles), linker disorder, guest arrangement and host-guest interactions can be attained with atomic precision by SXRD. These characteristics are impossible to be directly determined when only the polycrystalline samples are available. (*Science*,

2018, **361**, 48-52). Fortunately, we obtained the high-quality single crystal of the activated framework (8PN-Heated) from a high-quality single crystal of 8PN-EA, which is amenable to single-crystal X-ray diffraction characterization, therefore achieving all single crystal to single crystal transformations during solvent removal, which is not common for HOFs. From this SCSC transformation, we can directly and precisely observe that two dihedral angles between the plane of the ethylene core and the planes of two different phenyl rings in one arm of TPE-4pn change from 44.72° and 75.23° to 36.39° and 6.12° due to rotations of the phenyl rings, leading to the larger pore size of $6.507 \times 12.290 \text{ \AA}^2$ in 8PN-EA to the smaller pore size of $6.232 \times 9.624 \text{ \AA}^2$ in 8PN-Heated. This is a direct and strong evidence for the general rule promoted in our manuscript that the larger dihedral angles between the plane of the ethylene core and the plane of the phenyl ring can generally lead to the larger pore size.

Meanwhile, among the ten single crystals of 8PN suitable for the SXRD analysis, a large adjustable range of void space can be achieved (from 4.4% of total cell volume to 33.2% of total cell volume). As mentioned above, due to the exceptional flexibility of 8PN, multimode reversible structural transformations can be realized in these frameworks. More importantly, due to the tunable pore size and pore shape resulted from the exceptional flexibility of 8PN, the guests of varying shapes, sizes, aggregation states and even amounts can all be encapsulated by 8PN. All the co-crystals reported in our manuscript are suitable for the SXRD analysis. As a result, guest amount, guest arrangement and host-guest interactions can all be clearly determined. We believe 8PN can accommodate many special-purpose guests to create novel host-guest materials with some unexpected properties and functions. The employment of a flexible host in these host-guest co-crystals should also provide a new strategy for the conventional co-crystal preparations. In a word, the flexible 8PN can serve as a multifunctional platform, which should promote the development of porous materials.

To make the manuscript clearer, related description about the SCSC transformation has been added and our changes made to the revised manuscript are highlighted in yellow as follows:

On page 9 in the revised manuscript:

In order to get that single-crystal structure, a single crystal of 8PN-EA was heated carefully until all the EA molecules were removed to yield a new single crystal named

8PN-Heated. Upon the solvent removal, the single crystallinity of 8PN can be retained.

On page 10 in the revised manuscript:

After EA removal, two dihedral angles between the plane of the ethylene core and the planes of two different phenyl rings in one arm of TPE-4pn change from 44.72° and 75.23° to 36.39° and 6.12° due to rotations of the phenyl rings (Fig.3b), leading to the transformation of the large pore (LP) form to the narrow pore (NP) form. Therefore, the aforementioned general rule between dihedral angles and pore sizes is further clearly confirmed.

Figure 3 | Single-crystal-to-single-crystal (SCSC) transformation determined by the single-crystal X-ray diffraction. a, Structural transformations of the large pore (LP) form in 8PN-EA to the narrow pore (NP) form in 8PN-Heated accompanied by removal of solvent molecules. b, Illustration of the significant changes in the dihedral angles between the plane of the ethylene core and the planes of two different phenyl rings in one arm of TPE-4pn in 8PN-EA and 8PN-Heated due to phenyl ring-rotation.

REVIEWERS' COMMENTS:

Reviewer #1 (Remarks to the Author):

The authors have well addressed the reviewers' comments

Reviewer #2 (Remarks to the Author):

The authors have figured out all the questions. Their paper should be accepted by Nature Communactions.

Reviewer #3 (Remarks to the Author):

After considering the author's response and the other referees' comments, I can accept this manuscript's publication in Nat. Commun now.

Point-by point response to reviewers' comments and editorial requests:

(Reviewers' comments: in black; Responses to the comments: in blue)

Reviewer #1 (Remarks to the Author):

The authors have well addressed the reviewers' comments.

Response: We would like to thank you very much for your helpful suggestions and positive comments to polish our paper.

Reviewer #2 (Remarks to the Author):

The authors have figured out all the questions. Their paper should be accepted by Nature Communactions.

Response: We would like to thank you very much for your valuable comments on our manuscript.

Reviewer #3 (Remarks to the Author):

After considering the author's response and the other referees' comments, I can accept this manuscript's publication in Nat. Commun now.

Response: We would like to thank you very much for your recognition of our work and valuable comments.